# Fast Bayesian Coresets via Subsampling and Quasi-Newton Refinement

**Cian Naik**
Department of Statistics
University of Oxford
cian.naik@stats.ox.ac.uk

**Judith Rousseau**
Department of Statistics
University of Oxford
judith.rousseau@stats.ox.ac.uk

**Trevor Campbell**
Department of Statistics
University of British Columbia
trevor@stat.ubc.ca

## Abstract

Bayesian coresets approximate a posterior distribution by building a small weighted subset of the data points. Any inference procedure that is too computationally expensive to be run on the full posterior can instead be run inexpensively on the coreset, with results that approximate those on the full data. However, current approaches are limited by either a significant run-time or the need for the user to specify a low-cost approximation to the full posterior. We propose a Bayesian coreset construction algorithm that first selects a uniformly random subset of data, and then optimizes the weights using a novel quasi-Newton method. Our algorithm is a simple to implement, black-box method, that does not require the user to specify a low-cost posterior approximation. It is the first to come with a general high-probability bound on the KL divergence of the output coreset posterior. Experiments demonstrate that our method provides significant improvements in coreset quality against alternatives with comparable construction times, with far less storage cost and user input required.

## 1 Introduction

Bayesian methods are key tools for parameter estimation and uncertainty quantification, but exact inference is rarely possible for complex models. Currently, the gold standard method for approximate inference is Markov chain Monte Carlo (MCMC) [1; 2; 3, Ch. 11,12], which involves simulating a Markov chain whose stationary distribution is the Bayesian posterior. However, modern applications are often concerned with very large datasets; in this setting, MCMC methods typically have a $\Theta(NT)$ complexity—for $T$ samples and dataset size $N$ —which quickly becomes intractable as $N$ increases. Motivated by stochastic methods in variational inference [4, 5], this cost can be reduced by involving only a random subsample of $M \ll N$ data points in each Markov chain iteration [6–13]. This reduces the per-iteration computation time of MCMC, but it can create substantial error in the stationary distribution of the resulting Markov chain and cause slow mixing [11, 14–17].

Bayesian coresets [18–23] provide an alternative for reducing the cost of MCMC and other inference methods. The key idea is that in a large-scale data setting, much of the data is often redundant. In particular, there often exists a *fixed* small, weighted subset of the data—a *coreset*—that suffices to capture the dataset as a whole in some sense [21]. Thus, if one can find such a coreset, it can be used in place of the full dataset in the MCMC algorithm, providing the simplicity, generality, and per-iteration speed of data-subsampled MCMC without the statistical drawbacks. Current state-of-

the-art Bayesian coreset approaches based on sparse variational inference [21] empirically provide high-quality coresets, but are limited by their significant run-time and lack of theoretical convergence guarantees. Methods based on sparse regression [19, 20, 23] are significantly faster and come with some limited theoretical guarantees, but require a low-cost approximation to the full posterior. This approximation is often impractical to find and difficult to tune. Moreover, it can fundamentally limit the quality of the coreset obtained. Aside from [21], these methods also require $\Theta(N)$ storage, which is problematic in the large-data setting. There is a third class of constructions based on importance sampling [18], but these do not provide reliable posterior approximations in practice.

In this work, we develop a novel coreset construction algorithm that is both faster and easier to tune than the current state of the art, and has theoretical guarantees on the quality of its output. The key insight in our work is that we can split the construction of a coreset into two stages: first we select the data in the coreset via uniform subsampling—an idea developed concurrently by [24, 25]—and then optimize the weights using a few steps of a novel quasi-Newton method. Selecting points uniformly randomly avoids the slow inner-outer loop optimization of [21], and guarantees that the optimally-weighted coreset has a low KL divergence to the posterior (Theorems 4.1 and 4.2). Weighting these uniformly selected points correctly is crucial, and our quasi-Newton method is guaranteed to converge to a point close to the optimally-weighted coreset (Theorem 4.3) in significantly fewer iterations than the method of [21]. Finally, our method is easy to tune and does not require the low-cost posterior approximation of [19, 20]. Our experiments show that the algorithm exhibits significant improvements in coreset quality against alternatives with comparable construction times.

## 2  Bayesian coresets

The problem we study is as follows. Suppose we are given a target probability density $\pi(\theta)$ for variable $\theta \in \Theta$ that is comprised of $N$ potentials $(f_n(\theta))_{n=1}^N$ and base density $\pi_0(\theta)$,

$$\pi(\theta) := \frac{1}{Z} \exp \left( \sum_{n=1}^N f_n(\theta) \right) \pi_0(\theta), \tag{1}$$

where $Z$ is the normalizing constant. This setup corresponds to a Bayesian statistical model with prior $\pi_0$ and i.i.d. data $X_n$ conditioned on $\theta$, where $f_n(\theta) = \log p(X_n|\theta)$. Computing expectations under $\pi$ exactly is often intractable. MCMC methods can be employed here, but typically have a computational complexity of $\Theta(NT)$ to obtain $T$ samples, since $\sum_n f_n(\theta)$ needs to be evaluated at each step. Bayesian coresets [18] offer an alternative, by constructing a small, weighted subset of the dataset, on which any MCMC algorithm can then be run.

To do this, we find a set of weights $w \in \mathbb{R}_{\geq 0}^N$ corresponding to each data point, with the constraint that only $M \ll N$ of these are nonzero, i.e. $\|w\|_0 := \sum_{n=1}^N \mathbb{1}_{w_n > 0} \leq M$. The weighted subset of points corresponding to the strictly positive weights is called a *coreset*, and we can then create the coreset posterior approximation,

$$\pi_w(\theta) := \frac{1}{Z(w)} \exp \left( \sum_{n=1}^N w_n f_n(\theta) \right) \pi_0(\theta), \tag{2}$$

where $Z(w)$ is the new normalizing constant, and $\pi_1 = \pi$ corresponds to the full posterior. Running MCMC on this approximation has complexity $\Theta(MT)$, a considerable speedup if $M \ll N$.

Constructing a sparse set of weights $w$ so that $\pi_w$ is as close to $\pi$ as possible is the key challenge here. Methods based on importance sampling [18] and sparse regression [19, 20, 23] have been developed, but these provide poor posterior approximations or require first finding a low-cost posterior approximation. The current state-of-the-art approach resolves both of the aforementioned issues by formulating the coreset construction problem as variational inference in the family of coresets [21]:

$$w^\star = \underset{w \in \mathbb{R}^N}{\arg\min} \ \ \mathrm{D_{KL}} \left( \pi_w \| \pi_1 \right) \quad \text{s.t.} \quad w \geq 0, \|w\|_0 \leq M, \tag{3}$$

where $\mathrm{D_{KL}}(\cdot\|\cdot)$ denotes the KL divergence. Noting that coresets are a sparse subset of an exponential family—the weights form the natural parameter $w \in \mathbb{R}_{\geq 0}^N$, the component potentials $(f_n(\theta))_{n=1}^N$ form the sufficient statistic, $\log Z(w)$ is the log partition function, and $\pi_0$ is the base density,

$$\pi_w(\theta) := \exp \left( w^T f(\theta) - \log Z(w) \right) \pi_0(\theta), \quad f(\theta) := (f_1(\theta), \ldots, f_N(\theta))^T \in \mathbb{R}^N, \tag{4}$$

one can obtain a formula for the gradient

$$\nabla_w D_{KL} \left( \pi_w \| \pi_1 \right) = - \nabla_w^2 \log Z(w)(1 - w) = - \mathrm{Cov}_w \left[ f, f^T(1 - w) \right],$$ (5)

where $\mathrm{Cov}_w$ denotes covariance under $\pi_w$. Here, and throughout, we take derivatives with respect to $w$ on the ambient space $w \in \mathbb{R}^N$. The method of [21] then involves selecting a data point to add to the coreset, optimizing the coreset weights using stochastic gradient estimates based on Eq. (5), and then iterating. In practice, this method is infeasibly slow outside of small-data problems. For every data point selected it requires sampling from $\pi_w$ a number of times equal to the number of optimization steps (set to 100 in [21]). This can easily lead to tens of thousands of sampling steps in total, each of which typically necessitates the use of an MCMC algorithm. Conversely, our method requires tens of sampling steps in total.

## 3 Construction via subsampling and quasi-Newton refinement

In this section, we provide a new Bayesian coreset construction algorithm (Algorithm 1) to solve the sparse variational inference problem Eq. (3). This method involves first uniformly subsampling the data, and then optimizing the weights on the subsample. Here, we give a detailed explanation of how the algorithm works. We defer to Section 4 the theoretical analysis of the method, which demonstrates that, with high probability, it finds a near-optimal coreset in a small number of optimization iterations.

### 3.1 Uniform subsampling

The first key insight to our approach is that we can select the $M$ data points that comprise the coreset by simple uniform subsampling. From an algorithmic standpoint, the benefit of this approach is clear: it makes the selection of coreset points fast and easy to implement. It also decouples the subset selection from the optimizing of the coreset weights, leaving us with a simpler optimization problem to solve.[1] Intuitively, this is reasonable from a coreset quality standpoint because a uniform subsample of the data will, with high probability, create a highly-expressive "basis" of log-likelihood functions $(f_n)_{n=1}^M$ with which to approximate the full data log-likelihood. Of course, we still need to optimize the coreset weights; but we have not limited the flexibility of the coreset family significantly by selecting points via uniform subsampling. Theorem 4.2 provides a precise statement of these ideas, and shows that the optimal weighted coreset posterior built using a uniformly random subset of data is a good posterior approximation.

Since the coreset data points are chosen uniformly randomly, we will w.l.o.g. re-label the selected subset to have indices $1, \ldots, M$, and the remaining data points to have indices $M+1, \ldots, N$ from now on. We can specify a new weight constraint set $\mathcal{W} \subset \mathbb{R}^N$ of vectors $w$ with 0 entries for every index beyond $M$, and a further subset $\mathcal{W}_N$ where the weights sum to $N$,

$$\mathcal{W} = \left\{ w \in \mathbb{R}^N : w \geq 0, \ n > M \implies w_n = 0 \right\}, \qquad \mathcal{W}_N = \left\{ w \in \mathcal{W} : 1^T w = N \right\}.$$

It is also useful to denote $g(\theta) = (f_1(\theta), \ldots, f_M(\theta))^T \in \mathbb{R}^M$ as the first $M$ components of $f(\theta)$.

### 3.2 Quasi-Newton optimization

Given the choice of $M$ data points to include in the coreset, it is crucial to optimize their weights correctly. This is the second step of our algorithm. The KL divergence gradient for the $M$ weights is

$$\nabla_w D_{KL} \left( \pi_w \| \pi_1 \right) = \left( -\nabla_w^2 \log Z(w)(1 - w) \right)_{1:M} = - \mathrm{Cov}_w \left[ g, f^T(1 - w) \right].$$ (6)

To reduce the number of weight optimization iterations —each of which generally involves MCMC sampling from $\pi_w$, where $w$ is the current set of weights—we develop a second-order optimization method. Again using the properties of exponential families, we can derive the Hessian of the KL divergence with respect to the $M$ weights

$$\nabla_w^2 D_{KL} \left( \pi_w \| \pi_1 \right) = \left( \nabla_w^2 \log Z(w) - \nabla_w^3 \log Z(w)(1 - w) \right)_{1:M,1:M}.$$

---

[1]The benefits of this initial uniformly random selection step are also noted by [24, 25] in concurrent work.

This matrix is not guaranteed to be positive (semi-)definite. However, writing out the second term explicitly as

$$\nabla_w^3 \log Z(w)(1-w) = \mathbb{E}_{\pi_w}\left[(f - \mathbb{E}_{\pi_w}(f))(f - \mathbb{E}_{\pi_w}(f))^T\left((f^T(1-w) - \mathbb{E}_{\pi_w}(f^T(1-w)))\right)\right],$$

we see that this term will be small near the optimum, where the coreset approximation is ideally good, i.e., $w^T f(\theta) \approx 1^T f(\theta)$. The first term does not contain this $f^T(1-w)$ term, and thus should dominate the expression.

This motivates the use of

$$\nabla_w^2 D_{\mathrm{KL}}(\pi_w \| \pi_1) \approx \left(\nabla_w^2 \log Z(w)\right)_{1:M,1:M} = \mathrm{Cov}_w[g,g] \tag{7}$$

to scale gradient steps in the optimization method rather than the true Hessian, thus creating a quasi-Newton method [26]. This heuristic motivates our approach, but it is rigorously justified by Theorem 4.3, which proves that this scaling results in optimization iterations that are guaranteed to converge exponentially to a coreset which is nearly globally optimal. Note that when $M$ is larger than the inherent dimension of the space of log-likelihood functions, $\mathrm{Cov}_w[g,g]$ will have zero eigenvalues and be noninvertible. Therefore we add a regularization $\tau > 0$ prior to inversion. Theorem 4.3 shows how the regularization influences the optimization, but in general we want $\tau$ to be as small as possible while still ensuring the numerical stability of inverting $\mathrm{Cov}_w[g,g]$.

Our optimization method is as follows. First, we initialize at the uniformly weighted coreset, i.e.,

$$w_0 \in \mathcal{W}, \quad w_{0m} = \frac{N}{M}, \quad m = 1, \dots, M.$$

Next, given a step-size tuning sequence $\gamma_k$, $k \geq 0$, we update the weights at each iteration $k \in \mathbb{N} \cup \{0\}$ using the ($\tau > 0$)-regularized quasi-Newton step

$$\hat{w}_{k+1} = w_k + \gamma_k(G(w_k) + \tau I)^{-1}H(w_k)(1 - w_k) \tag{8}$$

$$G(w) = \mathrm{Cov}_w[g,g], \quad H(w) = \mathrm{Cov}_w[g,f]. \tag{9}$$

Theorem 4.3 suggests that the step size $\gamma_k$ should be constant: $\gamma_k = \gamma \in (0,1]$; but the analysis in that section assumes that we can compute $G(w)$ and $H(w)$ exactly, whereas in practice we will have to estimate them. Hence we allow $\gamma_k$ to depend on the iteration number in general. Finally, we project $\hat{w}_{k+1}$ back onto the constraint set to obtain next iterate $w_{k+1}$, i.e. for $m = 1, \dots, M$ set

$$w_{k+1,m} = \max(\hat{w}_{k+1,m}, 0). \tag{10}$$

### 3.3 Algorithm

The pseudocode for the quasi-Newton coreset construction method is shown in Algorithm 1. There are a number of practical considerations needed to use this method; we discuss these here.

The first consideration is that we cannot calculate $G(w)$ and $H(w)$ in closed form. Instead, at step $k$ of the algorithm, we use a Monte Carlo estimate; first taking $S$ samples $(\theta_s)_{s=1}^S \overset{\text{i.i.d.}}{\sim} \pi_{w_k}$, and calculating $\hat{g}_s := g(\theta_s) - \frac{1}{S}\sum_{r=1}^S g(\theta_r)$, so that we may estimate $G(w_k)$ as

$$\hat{G}_k = \frac{1}{S}\sum_{s=1}^S \hat{g}_s \hat{g}_s^T \in \mathbb{R}^{M \times M}. \tag{11}$$

Using a Monte Carlo estimate in this way introduces one source of error. Furthermore, we often cannot sample exactly from $\pi_{w_k}$, and instead use MCMC to do so. This could lead to additional errors, for example if our samples are from unconverged chains. However, we find that this method works well in practice, for a reasonable number of samples $S$.

As $H(w_k)$ is an $M \times N$ matrix, estimating it directly will incur a $\Theta(MN)$ storage cost. Instead, we note that we only require $H(w_k)(1 - w_k) = \mathrm{Cov}_{w_k}[g, f^T(1-w_k)] = \mathrm{Cov}_{w_k}[g,h]$, where $h := f^T 1 - g^T w_k = \bar{f} - g^T w_k$. We can estimate $h$ using $\hat{h}_s = \sum_{n=1}^N\left[f_n(\theta_s) - \frac{1}{S}\sum_{r=1}^S f_n(\theta_r)\right] - \hat{g}_s^T w_k$, which requires $O(N)$ time, but only $O(S)$ space. We can then estimate $H(w_k)(1 - w_k)$ as

$$\hat{H}_k(1 - w_k) = \frac{1}{S}\sum_{s=1}^S \hat{g}_s \hat{h}_s. \tag{12}$$

---

**Algorithm 1** QNC (QUASI-NEWTON CORESET)

---

**Require:** $(f_n(\theta))_{n=1}^N$ $\pi_0$, $S$, $K$, $K_{tune}$, $\gamma$, $\tau$, $M$.
 1: Uniformly sample $M$ data points, w.l.o.g. these correspond to indices $1, \ldots, M$.
 2: Set $w_{0m} = \frac{N}{M}, \quad m = 1, \ldots, M$.
 3: **for** $k = 0, \ldots, K-1$ **do**
 4:      Sample $(\theta_s)_{s=1}^S \stackrel{\text{i.i.d.}}{\sim} \pi_{w_k} \propto \exp\left(w_k^T f(\theta)\right) \pi_0(\theta)$
 5:      Set $\hat{g}_s \leftarrow g(\theta_s) - \frac{1}{S} \sum_{r=1}^S g(\theta_r)$ and $\hat{h}_s \leftarrow \sum_{n=1}^N \left[ f_n(\theta_s) - \frac{1}{S} \sum_{r=1}^S f_n(\theta_r) \right] - \hat{g}_s^T w_k$
 6:      Set $\hat{H}_k(1 - w_k) \leftarrow \frac{1}{S} \sum_{s=1}^S \hat{g}_s \hat{h}_s$ and $\hat{G}_k \leftarrow \frac{1}{S} \sum_{s=1}^S \hat{g}_s \hat{g}_s^T$
 7:      **if** $k \le K_{\text{tune}}$ **then**
 8:         Choose $\gamma_k$ via line search with starting value $\gamma$
 9:      **else**
10:         Set $\gamma_k \leftarrow \gamma$
11:      Take a quasi-Newton step: $\hat{w}_{k+1} \leftarrow w_k + \gamma_k \left(\hat{G}_k + \tau I\right)^{-1} \hat{H}_k(1 - w_k)$
12:      Project: for all $m \in [M]$, $w_{k+1,m} \leftarrow \max\left(\hat{w}_{k+1,m}, 0\right)$
13: **return** $w = w_K$

---

Thus, we can find $w_{k+1}$ from $w_k$ by taking the stochastic Newton step

$$\hat{w}_{k+1} = w_k + \gamma_k \left(\hat{G}_k + \tau I\right)^{-1} \hat{H}_k(1 - w_k), \tag{13}$$

and projecting onto the constraint set: $w_{k+1,m} = \hat{w}_{k+1,m} \mathbb{1}_{\hat{w}_{k+1,m} \ge 0}$. We set the regularization parameter $\tau$ by examining the condition number of $\hat{G}_k + \tau I$ and keeping it below a reasonable value. We can tune $\gamma_k$ using a line search method. As we do not have access to the objective function (i.e. the KL-divergence between the coreset and full posteriors), we use the curvature part of the Wolfe conditions [27] to tune this. In practice, this line search is expensive. Thus, we only tune $\gamma_k$ for $k \le K_{\text{tune}}$, and leave it as a constant thereafter. The intuition here is that we may start with quite poor coreset weights, and so need to choose the initial steps carefully. In Appendix C, we perform a sensitivity analysis for the parameters $S$, $K_{tune}$ and $\tau$. We see that our results are generally not sensitive to the choice of these parameters, within reasonable ranges.

Computing $\hat{G}_k + \tau I$ involves taking $S$ samples, forming the product $\hat{g}_s \hat{g}_s^T$, and then inverting the resulting matrix. Its time complexity is thus $O(SM^2 + M^3)$. Computing $\hat{H}(w_k)(1 - w_k)$ similarly has complexity $O(SM + SN)$, and computing the product with $(\hat{G}_k + \tau I)^{-1}$ has complexity $O(M^2)$. For $K$ Newton steps in Algorithm 1, the overall time complexity is thus $O(K(M^3 + SM^2 + SN))$, which is linear in $N$. The space complexity is $O(M^2)$ (to store $\hat{G}_k$), which is sublinear in $N$, unlike in [20, 23].

In theory, we can reduce the time complexity by using a further stochastic estimate of $\hat{H}_k(1 - w_k)$. In particular, we can instead calculate $\hat{h}_s$ only for indices in $\mathcal{I} := \{1, \ldots, M\} \cup \mathcal{T}$, where $\mathcal{T}$ is a uniformly selected sample $\mathcal{T} \subseteq \{1, \ldots, N\}$. Calculating $\hat{H}(w_k)(1 - w_k)$ using this subsampled vector $\hat{h}_{s,\mathcal{I}}$ then has complexity $O(SM + ST)$ in the worst case, where $T := |\mathcal{T}|$. The overall time complexity is then $O(K(M^3 + SM^2 + ST))$, which is sublinear in $N$ if $T = o(N)$. This could give significant improvements in coreset construction time in the large-data regime. However, this approach requires further study, as we find that it leads to a degradation in the performance of our algorithm. We do not use it in our experiments.

The final practical consideration in the design of Algorithm 1 is that we cannot actually calculate the objective function that we are trying to minimize, namely the KL divergence $D_{\text{KL}}(\pi_w \| \pi_1)$ between the coreset and full posteriors. Thus, it may be hard to tell if the optimization is actually making progress. In practice, we monitor the norm of the gradient of the KL divergence (which we do have access to, as given by Eq. (6)). We can terminate our algorithm early if we do not see a significant enough decrease in this measure.

# 4 Theoretical analysis

In this section we provide a theoretical analysis of the proposed method. We demonstrate (in Theorems 4.1 and 4.2) that the optimal coreset posterior built using a uniformly random subset of data is, with high probability, an exact or near-exact approximation of the true posterior. Second, we show (in Theorem 4.3) that the proposed optimization algorithm converges exponentially quickly to a point near the optimal coreset, thus realizing the guarantee in Theorem 4.2 in practice, and theoretically justifying our proposed algorithm. The assumptions required for our theory are somewhat technical. We give some intuition here on settings where they hold, but discuss this further in Appendix A. Proofs of all results may be found in Appendix B.

## 4.1 Subset selection via uniform subsampling

Theorem 4.1 and Theorem 4.2 provide a theoretical foundation for selection via uniform subsampling. Both recommend setting $M \gtrsim d \log N$, where $d$ is essentially the "dimension" of the span of the log-likelihood functions. Theorem 4.1 states that if the span of the log-likelihood functions is indeed finite-dimensional with dimension $d$—e.g., in a usual exponential family model—then with high probability, the optimal coreset using a uniformly random selection of $M$ points is exact. Theorem 4.2 extends this to the general setting where $\Theta \subseteq \mathbb{R}^D$, $N$ is large relative to $D$, and where we assume the posterior concentrates as $N \to \infty$. In particular, Theorem 4.2 states that with high probability, the optimal coreset using a uniformly random selection of $M$ data points provides low error.

In what follows, we assume that $f_n(\cdot)$ are i.i.d. from some true data-generating distribution $p$, and we define $\bar{f}(\cdot) = \mathbb{E}_p(f_n(\cdot))$. We also introduce the new notation $f(\cdot, \cdot)$, where we consider the potentials $f$ as functions of both $x$ and $\theta$. This enables us to make the data-dependence explicit when needed, i.e., $f_n(\cdot) = f(x_n, \cdot)$; for instance, in the standard context of Bayesian inference $f_n(\cdot) = f(x_n, \cdot) = \log p(x_n|\cdot)$, with $x_n \overset{\text{i.i.d.}}{\sim} p_0$ for $n \in [N]$.

**Theorem 4.1.** *Let $S_0$ be the vector space of functions on $\Theta$ spanned by $\{f(x, \cdot); x \in \mathcal{X}\}$. Assume that $S_0$ is finite dimensional, with dimension $d$. Let $\mu$ be any measure on $\Theta$ such that $S_0 \subset L^2(\mu)$ and denote $r_\mu^2 = \mathbb{E}_\mu \left( \mathbb{E}_p([f_1 - \bar{f}]^2) \right)$. For $\delta > 0$ define*

$$J(\delta) = \inf_{a \in S_0; \|a\|=1} \Pr \left( \langle a, f_1 - \bar{f} \rangle_{L^2(\mu)} > \frac{2r_\mu}{\sqrt{N\delta}} \right). \tag{14}$$

*Then for $\delta > 0$ such that $J(\delta) + \delta < 1$, a universal constant $C_1$, and $M \in \mathbb{N}$ such that*

$$M \geq \frac{2d}{J(\delta)} \left( \log N + \log \left( \frac{4\delta}{J(\delta)} \right) + C_1 \right),$$

*we have that with probability $\geq 1 - \delta - e^{-MJ(\delta)/4}$,*

$$\min_{w \in \mathcal{W}_N} \mathrm{KL}(\pi_w \| \pi) = 0.$$

Note that the condition that $S_0$ has finite dimension $d < \infty$ is satisfied for all $d$-dimensional exponential families when $f_n(\theta) = \log p(x_n|\theta)$. Further, note that Theorem 4.1 still applies in situations where $d$ increases with $N$. To understand the behaviour of $J(\delta)$, we consider in Appendix A a full rank exponential family. We can show that, under certain conditions, $J(\delta)$ converges to a strictly positive value as $\delta N \to \infty$. In this case, we can set $\delta = \omega(1/N)$, such that with probability $\geq 1 - \omega(1/N)$, we indeed have that $\min_{w \in \mathcal{W}_N} \mathrm{KL}(\pi_w \| \pi) = 0$ for $M \gtrsim d \log N$.

In the interest of obtaining a more general approximation result, we now consider the case where $S_0$ is infinite dimensional, but where we can find a finite-dimensional space $S_1$ which approximates $f_n(\theta) - \bar{f}(\theta)$ reasonably well. A typical setup where this is valid is when $\Theta \subseteq \mathbb{R}^D$, the posterior concentrates (as $N \to \infty$) at a point $\theta_0 \in \Theta$ that maximizes $\bar{f}(\theta)$, and each $f_n(\cdot)$ is smooth near $\theta_0$ and globally well-behaved in some sense. Theorem 4.2 below states, roughly, that for a model satisfying these assumptions, $\min_{w \in \mathcal{W}_N} \mathrm{KL}(\pi_w \| \pi)$ will be small with high probability. We give the exact conditions in Appendix A. Although they are perhaps strong, the result is still indicative of when a good posterior approximation is obtained by our method.

Intuitively, we require that $f(x, \theta)$ be differentiable, strongly concave in $\theta$, and "smooth enough". Letting $\theta_0$ be the unique value such that $\bar{f}(\theta) \leq \bar{f}(\theta_0) \ \forall \theta \in \Theta$, we also require that we can split

$f(x; \theta)$ into two parts,

$$f(x, \theta) = f^{(1)}(x, \theta) + f^{(2)}(x, \theta),$$

where the span of $\{f^{(1)}(x, \cdot) - f^{(1)}(x, \theta_0) : x \in \mathcal{X}\}$ is a finite-dimensional vector space $S_1$. This split needs to be made such that $f^{(2)}$ is negligible, in the sense that $f(x, \theta) \approx f^{(1)}(x, \theta)$ locally around $\theta_0$. A usual case where these conditions hold is when $f(x; \theta)$ is $C^3$ in $\theta$. In this case, we can obtain $f^{(1)}(x; \theta)$ by a second-order Taylor expansion,

$$f^{(1)}(x; \theta) = f(x; \theta_0) + (\theta - \theta_0)^T \nabla_\theta f(x; \theta_0) + (\theta - \theta_0)^T \nabla^2 f(x; \theta_0)(\theta - \theta_0)/2,$$

which has dimension bounded by $1 + D + D(D+1)/2$. $f^{(2)}(x; \theta)$ is then the remainder term, which is indeed negligible around $\theta_0$ in the desired sense. In Appendix A, we provide further discussion on the conditions of Theorem 4.2. Recalling that a quantity is $o(1)$ as $N \to \infty$ if and only if it converges to 0, we can then state the result as follows.

**Theorem 4.2.** *Suppose the assumptions A1-A4 in Appendix A hold, and we set $M \gtrsim D(\log N + 1)$. Then as $N \to \infty$, with probability at least $1 - o(1)$, we have that*

$$\min_{w \in \mathcal{W}_N} \mathrm{KL}(\pi_w || \pi) = o(1).$$

In the above example, we can examine the proof to obtain a bound on $\mathrm{KL}(\pi_w || \pi)$ of order $1/\sqrt{N}$ with probability converging to 1, showing that this result does indeed lead to a useful bound.

## 4.2 Quasi-Newton weight optimization

Theorems 4.1 and 4.2 guarantee the existence of a high-quality coreset, but do not provide any insight into whether it is possible to find it tractably. Theorem 4.3 addresses this remaining gap. In particular, intuitively, it asserts that as long as (1) the regularization parameter $\tau > 0$ is nonzero but small enough that it does not interfere with the optimization (in particular, the minimum eigenvalue of $G(w)$ over the space), and (2) the *optimal* coreset is a good approximation to the full dataset (which is already guaranteed by Theorem 4.2), then the weights $w_k$ at iteration $k$ of the approximate Newton method converge exponentially to a point close to a global optimum. In the result below, let $G(w)$ and $H(w)$ be defined as in Eq. (9).

**Theorem 4.3.** *Let $W \subseteq \mathcal{W}$ be a closed convex set, $W^\star \subseteq \mathrm{argmin}_{w \in W} \mathrm{KL}(\pi_w || \pi)$ be a maximal closed convex subset, and fix the regularization parameter $\tau > 0$. Define*

$$\xi = \inf_{w \in W} \min \left\{ \frac{\lambda}{\lambda + \tau} : \lambda \in \mathrm{eigvals}\, G(w), \lambda > 0 \right\}.$$

*Suppose $\exists\, \epsilon \in [0, 1)$ and $\delta \geq 0$ such that for all $w \in W$*

$$\| (G(w) + \tau I)^{-1} H(w)(1 - w^\star) \| \leq \epsilon \|w - w^\star\| + \delta, \tag{15}$$

*where $w^\star = \mathrm{proj}_{W^\star}(w)$. Then the $k^{th}$ iterate $w_k \in W$ and its projection $w_k^\star = \mathrm{proj}_{W^\star}(w_k)$ of the projected Newton method defined in Section 3.2 by Eqs. (8), (9) and (10), initialized at $w_0 \in W$ with a fixed step size $\gamma \in [0, 1]$ and regularization $\tau$ satisfies*

$$\|w_k - w_k^\star\| \leq \eta^k \|w_0 - w_0^\star\| + \gamma\delta \left( \frac{1 - \eta^k}{1 - \eta} \right), \qquad \eta = 1 - \gamma(\xi - \epsilon).$$

For this result to make sense, we require that $\eta \in [0, 1]$. We have that $\eta \geq 0$, since $\gamma \in [0, 1]$, $\epsilon \geq 0$ and $\xi \leq 1$ by definition. It may be the case that $\eta \geq 1$, in which case our result still holds, but is vacuous.

In order to use this result, the minimum positive eigenvalue of $G(w)$ for $w \in W \subseteq \mathcal{W}$ needs to be bounded, as does the error of the optimal coreset weights $w^\star$ in the sense of Eq. (15). A notable corollary of Theorem 4.3 occurs when there exists a $w^\star$ such that

$$\forall \theta \in \Theta, \quad \sum_{m=1}^{M} w_m^\star f_m(\theta) = \sum_n f_n(\theta),$$

i.e., the optimal coreset is identical to the true posterior distribution, as is guaranteed by Theorem 4.1. For any model where the conditions of Theorem 4.1 hold, such as any exponential family model, Eq. (15) holds with $\epsilon = \delta = 0$. We can thus take $\gamma = 1$ to find that

$$\|w_k - w_k^\star\| \leq (1 - \xi)^k \|w_0 - w_0^\star\|.$$

This provides some intuition on Theorem 4.3; as long as the optimal coreset posterior is a reasonable approximation to $\pi$, and we use a small regularization $\tau > 0$ (such that $\xi \approx 1$), then we generally need only a small number of optimization steps to find the optimal coreset. The conditions of Theorem 4.1 are sufficient, but not necessary, for Eq. (15) to hold. We present further discussion in Appendix A.

We note that if $\xi = 0$, then there is no convergence, even in the setting where Theorem 4.1 holds. However, it is important to emphasises that, for any given dataset, we are free to choose $\tau$. We can (and should) choose $\tau$ to be sufficiently small that $\xi$ is roughly 1. In particular, this means setting $\tau$ significantly smaller than the minimum positive eigenvalue of $G(w)$.

## 5    Experiments

In this section, we compare our proposed quasi-Newton coreset (QNC) construction against existing constructions—uniform subsampling (UNIF), greedy iterative geodesic ascent (GIGA) [20] and iterative hard thresholding (IHT) [23]—as well as the Laplace approximation (LAP), which represents what one obtains by assuming posterior normality in the large-data setting. Experiments were performed on a machine with a 2.6GHz 6-Core Intel Core i7 processor, and 16GB memory; code is available at `https://github.com/trevorcampbell/quasi-newton-coresets-experiments`.

In each case, we use $S = 500$ Monte Carlo samples during coreset construction. We compute error metrics using 1000 samples from each method's approximate posterior. We also compare to the baseline of sampling from the full posterior (FULL) to establish a noise floor for the given comparison sample size of 1000. In the synthetic Gaussian experiment, which is the simplest one we consider, we see that sparse variational inference (SVI) [21] is prohibitively slow, taking $4.2 \times 10^3$s to construct a coreset of size 1 (and scaling at best linearly with coreset size). Thus, we do not compare against it here. In Appendix C we run a smaller, lower dimensional, synthetic Gaussian experiment. Here, we compare against SVI, and confirm that it is prohibitively slow for the size of datasets that we consider.

For GIGA and IHT, we need to supply a low-cost approximation $\hat{\pi}$ to the posterior. To ensure these methods apply as generally as QNC and SVI, we use a uniformly sampled coreset approximation of size $M$ with weights $N/M$ (where $M$ is the same as the desired coreset size we are constructing). We also use these weights for UNIF. In Appendix C we provide additional results for GIGA and IHT with a Laplace approximation used for $\hat{\pi}$. This does not lead to a significant improvement, and we provide further discussion there.

Experiments are performed in settings standard in the Bayesian coresets literature, but with larger dataset sizes than those that could be considered previously. For each experiment we plot the approximate reverse KL divergence obtained by assuming posterior normality, and the build time. In Appendix C we provide comparisons on the additional metrics of relative mean and covariance error, forward KL divergence and per sample time to sample from the respective posteriors. For the experiments in Sections 5.2 and 5.3 with heavy tailed priors, we also compare using the maximum mean discrepancy [28] and kernel Stein discrepancy [29, 30], with inverse multi-quadratic (IMQ) kernel.

Throughout the experiments, we see that QNC outperforms the other subsampling methods for most coreset sizes we consider. GIGA and IHT in particular are limited by a fixed choice of $\hat{\pi}$ and finite projection dimension defined by our choice of $S = 500$—we discuss this further in Appendix C. Furthermore, outside of the synthetic Gaussian experiment, we see that QNC outperforms LAP for coreset sizes above certain thresholds, which represent a small fraction of the full dataset.

In Appendix C, we also perform a sensitivity analysis for the parameters $S$, $K_{tune}$ and $\tau$ that we use in Algorithm 1. We see that our results are generally not sensitive to the choice of these parameters, within reasonable ranges.

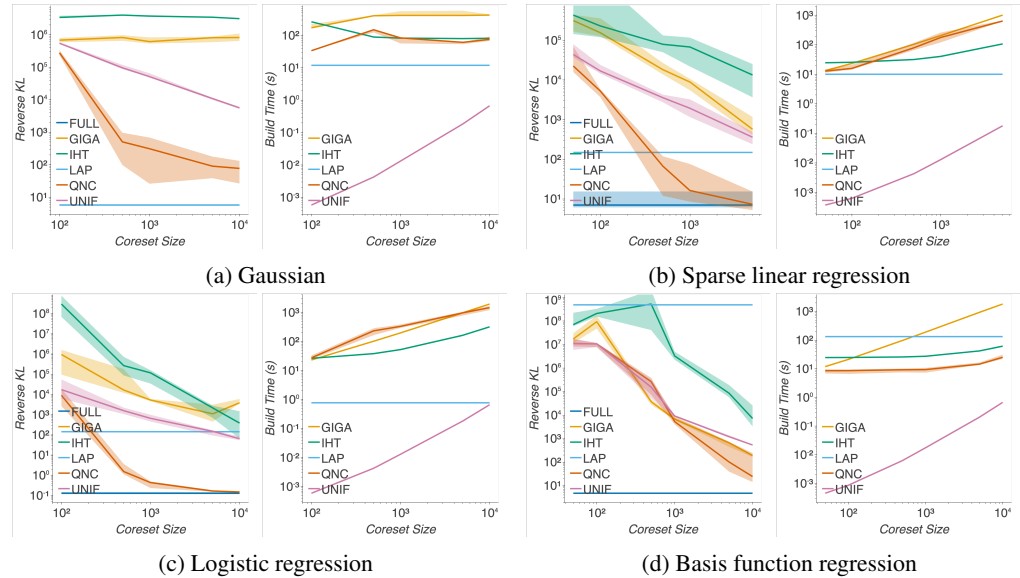

| (a) Gaussian | (b) Sparse linear regression |
| --- | --- |

| (c) Logistic regression | (d) Basis function regression |
| --- | --- |

Figure 1: Reverse KL divergence (left) and build time in seconds (right) for each experiment. We plot the median and a shaded area between the 25th/75th percentiles over 10 random trials. Our algorithm (QNC) provides an improvement in coreset quality, with a comparable run-time and less user input.

## 5.1 Synthetic Gaussian location model

Our first comparison is on a Gaussian location model, with prior $\theta \sim \mathcal{N}(\mu_0, \sigma_0 I)$, and likelihood $(X_n)_{n=1}^{N} \overset{\text{i.i.d.}}{\sim} \mathcal{N}(\theta, \sigma I)$ in $D$ dimensions. Here, we take $\mu_0 = 0$ and $\sigma_0 = 1$. Closed form expressions are available for the subsampled posterior distributions [21, Appendix B], and we can sample from them without MCMC. We compare the methods on a synthetic dataset with $N = 1,000,000$, $D = 100$, where we generate the $(X_n)_{n=1}^{N} \overset{\text{i.i.d.}}{\sim} \mathcal{N}(\mu, \sigma I)$ and set $(\mu_i)_{i=1}^{D} \overset{\text{i.i.d.}}{\sim} \mathcal{N}(0, 100)$ and $\sigma = 100$. From Fig. 1a we see that our method outperforms the other subsampling methods for all coreset sizes. This example is largely illustrative; we expect LAP to provide the exact posterior here by design, and this is indeed what we find (the reverse KL plots for LAP and FULL overlap).

## 5.2 Bayesian sparse linear regression

Next, we study a Bayesian sparse linear regression problem, where the data $(x_n, y_n)_{n=1}^{N}$ consists of a feature $x_n \in \mathbb{R}^D$ and an outcome $y_n \in \mathbb{R}^N$. The posterior distribution is that of $\theta \in \mathbb{R}^D$ in the model

$$y_n = x_n^T \theta + \epsilon_n, \quad n = 1, \dots, N \tag{16}$$

where $\epsilon_n \overset{\text{i.i.d.}}{\sim} \mathcal{N}(0, \sigma)$, $n = 1, \dots, N$. We place independent Cauchy priors on the coefficients $\theta$:

$$\theta_i \overset{\text{i.i.d.}}{\sim} \text{Cauchy}(0, \tau \lambda_i), \quad i = 1, \dots, D,$$

where the hyperpriors are $\lambda_i \overset{\text{i.i.d.}}{\sim} \text{Half-Cauchy}(0, 1)$, $\tau \sim \text{Half-Cauchy}(0, \sigma_0)$, with $\sigma_0 = 2$, and $\sigma \sim \Gamma(a_0, b_0)$ with $a_0 = 1, b_0 = 1$. We perform posterior inference on the $2D + 2-$dimensional set of parameters $(\theta_1, \dots, \theta_D, \lambda_1, \dots, \lambda_D, \sigma, \tau)$. Sampling is performed using STAN [31]. The dataset we study is a flight delays dataset,[2] with $N = 100,000$ and $D = 13$ (so that the overall inference problem is $28-$dimensional). The response variable $y_n$ is the delay in the departure time of a flight, and the features are meteorological and flight-specific information. We see in Fig. 1b that our method outperforms the other subsampling methods for all coreset sizes, and LAP for sizes above $500$—representing $0.5\%$ of the data. In Appendix C we see that the target posterior in this case has heavy tails, which makes the Laplace approximation particularly unsuited to this problem. We can see the effect this has even more clearly in the additional results presented there.

---

[2]This dataset was constructed by merging airport on-time data from the US Bureau of Transportation Statistics `https://www.transtats.bts.gov/DL_SelectFields.asp?gnoyr_VQ=FGJ` with historical weather records from `https://wunderground.com`.

### 5.3 Heavy-tailed Bayesian logistic regression

For this comparison, we perform Bayesian logistic regression with parameter $\theta \in \mathbb{R}^D$ having a heavy-tailed Cauchy prior $\theta_i \overset{\text{i.i.d.}}{\sim} \text{Cauchy}(0, \sigma)$, $i = 1, \ldots, D$. The data $(x_n, y_n)_{n=1}^{N}$ consists of a feature $x_n \in \mathbb{R}^D$ and a label $y_n \in \{-1, 1\}$. The relevant posterior distribution is that of $\theta \in \mathbb{R}^D$ which governs the generation of $y_n$ given $x_n$ via

$$y_n \mid x_n, \theta \overset{\text{indep}}{\sim} \text{Bern}\left(\frac{1}{1 + e^{-x_n^T \theta}}\right). \tag{17}$$

Again, sampling is performed using STAN. As in Section 5.2, we use the flight delays dataset, so we have that $N = 100,000$ and $D = 13$. The response variables $y_n$ are binarized, so that $y_n = 1$ if a flight was cancelled or delayed by more than an hour, and $y_n = -1$ if it was not. In Fig. 1c we see that our method outperforms other subsampling methods for all coreset sizes, and LAP for sizes above 500—representing $0.5\%$ of the data.

### 5.4 Bayesian radial basis function regression

Our final comparison is on a Bayesian basis function regression example. This is a larger version of the same experiment performed by [21], with $D = 301$ as before, but $N = 100,000$. Full details are given in Appendix C. From Fig. 1d we see that our method provides a significant improvement over LAP for all coreset sizes, which performs particularly poorly in this experiment, despite the Gaussianity of the target posterior [21, Appendix B]. It also outperforms the other subsampling methods for coreset sizes of 1000 and above (representing $1\%$ of the data) but this difference is not as marked as in the other examples. The fact that QNC requires a large coreset to start providing a benefit beyond UNIF stems from the fact that, in this higher dimensional example, there are sometimes data points that are very influential for a certain basis, and both UNIF and QNC can miss these. This indicates the need (in some settings) for a secondary method that can check for and include these, without the significant cost that e.g. SVI entails. We defer this study to future work. However, we do note that for this example QNC has the lowest build time, even outperforming LAP. This is because we only perform a small number of quasi-Newton steps before no further improvement is possible.

## 6 Conclusion

This paper introduces a novel method for data summarization prior to Bayesian inference. In particular, the method first selects a small subset of data uniformly randomly, and then optimizes the weights on those data points using a novel quasi-Newton method. Theoretical results demonstrate that the method is guaranteed to find a near-optimal coreset, and that the optimal coreset has a low KL divergence to the posterior with high probability. Future work includes investigating the performance of the method in more complex models, and studying conditions under which the method can provide compression in time sublinear in dataset size.

## Acknowledgments and Disclosure of Funding

T. Campbell was supported by a National Sciences and Engineering Research Council of Canada (NSERC) Discovery Grant and Discovery Launch Supplement. C. Naik was supported by the Engineering and Physical Sciences Research Council and Medical Research Council [award reference 1930478]. J. Rousseau was supported by the European Research Council (ERC) under the European Union's Horizon 2020 research and innovation programme (grant agreement No 834175).

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
