# OpenReview forum: "Fast Bayesian Coresets via Subsampling and Quasi-Newton Refinement"
_NeurIPS.cc/2022/Conference — NeurIPS 2022 Accept_

### Official Review · Reviewer_No1L · 2022-07-06

**Rating:** 7
**Confidence:** 3
**Soundness:** 3 good
**Presentation:** 3 good
**Contribution:** 3 good

**Summary:**

This paper proposes a method for approximating a posterior distribution in the framework of Bayesian statistics. Specifically, this posterior distribution involves a sum over $N$ potentials $\sum_{n=1}^N f_n(\theta)$ which has to be evaluated each time an expectation wrt the posterior has to be calculated. It is proposed to replaced this sum of another weighted sum over $M \ll N$ terms.  This corresponds to selecting a subset of data, called a coreset.
These points are sampled uniformly at random and their weights are computed by minimizing the KL divergence between the full posterior and the approximated posterior. A Quasi-Newton optimization method is proposed to calculate these weights.
Statistical guarantees are given for the size of the coreset in order to reach a small KL divergence. The convergence of the Quasi-Newton method is also discussed. Numerical simulations illustrate the accuracy of the posterior approximation.

**Questions:**

I have several questions about Thm 4.1 and its proof.
- First, the statement of this result is difficult to interpret since $J(\delta)$ depends on $\delta$ in a non-trivial way. After Thm 4.1, the authors try to explain the behaviour of $J(\delta)$ at the lines 185-191.
I do not find this explanation very clear. Although we might guess what are the random variable $X$ and $X_1$ from the context, it is worth defining them explicitly.
The limiting behaviour for $J(\delta)$ at line 189 is not obvious. Can you give a few lines of explanation?
- Second, it is difficult to follow the proof of Thm 4.1 in Supplementary Material. Could you give a roadmap for this proof? What are the main steps? It would help the reader to read this proof.

Here are a few technical questions about the proof of Thm 4.1.
- At line 61 (SM), the authors introduce a variable $t$, which is presumably in the $[0,1]$ interval. It is worth defining $t$.
Next, at line 62 (SM), there is a formula for the KL divergence. Can you give a few words of explanation? Why is this integral equal to the KL divergence?


- Also, at line 75 (SM), the authors claim they use a union bound. This step necessitates a clarification. Could you give a few details?
What are the events that you consider in the union? How do you bound $\phi$?

- At line 79 (SM), there is a typo. At the right-hand side of first inequality, there should be $\Pr( \dots < t_N)$ but $<t_N$ is missing. The same thing happen at the second inequality.

- At line 82 (SM), I presume there is another typo. In my opinion, $\log(A_d^{-d/2})$ should be replaced by $\log(A_d^{d/2})$. Also, the last equality should be an inequality.

Also, I have a few questions about the proof of Thm 4.2

- At line 98 (SM), the authors upper bound $I_{\epsilon, 1}$. The last inequality is unclear. I presume that the authors claim that
$[g - \sum_m v_m (f_m - \bar{f})-E_{w(v),t}]^2\leq [g - \sum_m v_m (f_m - \overline{f})]^2$.
Why is this correct?
Is the quantity $g - \sum_m v_m (f_m - \bar{f}) - E_{w(v),t}$ positive?

- At line 101 and at several other places (e.g. line 113), it is written that the probability of an event is bounded from below by $1 - f(N)$, or some decreasing function $f(N)$. To be complete, it should be mentioned that $N$ has to be large enough so that $f(N)<1$.

- At line 100 (SM), the authors write *We now assume $\|S_N(\theta)\| \leq (2- \sqrt{2}) \epsilon L_0 \sqrt{N}$*. This is not an assumption since the authors calculate right after the probability that this event does not happen. See also line 120 for a similar use of the verb *assume*.

- At line 102 (SM), another inequality is unclear. At the second lower bound, it seems that the authors take the square of an inequality. Are both sides of this inequality positive numbers?

- At line 127 (SM), the first equality is unclear. Could you give some details?

Here are some questions about Thm 4.3.

- At line 229, I think that the assumptions are incomplete, namely, about the parameter $\eta = 1 - \gamma(\xi - \epsilon)$ which influences the convergence rate. What are the conditions on these parameters such that $\eta >0$? Shouldn't there be conditions on $\epsilon$ so that $\xi - \epsilon >0$?

- About the convergence guarantees, there is an upper bound on $\| w_k - w_k^\star\|$ where $w_k^\star$ is the projection of $w_k$ on $W^\star$. Here, $w_k$ is the weight at iteration $k$. What does this bound tell about the convergence to a global optimum?


Typos:

104 unfinished sentence

Algorithm 1: indicies &larr; indices

SM 34 extra full stop

SM 87 $S_N$ &larr; $S_N(\theta_0)$

SM 122 missing ) on the left-hand side

=========================================================================

Note: I have read the authors' response which I found convincing. I update my score.


**Limitations:**

The limitations of this work are summarized in the checklist and are briefly discussed in section 5.4. I do not see any potential negative societal impact.

**Strengths And Weaknesses:**

Originality: This paper discusses interesting ideas. Being familiar with coresets in general but less with this specific application, I cannot completely assess the originality of this paper.

Quality: From a global perspective, the main paper is well-written and the results are well presented. However, from the technical viewpoint, the interpretation of some mathematical claims is not straightforward, for instance, in the case of Thm 4.1. The numerical simulations are convincing.

Clarity: The main paper is easy to follow. Concerning the technical details, I have several questions about the proofs of the theoretical results in Supplementary Material. These proofs are long (e.g., the proof of Thm 4.2 takes more than 8 pages). In principle, this is not an issue. However, these proves deserve extra clarifications at several places and are not easy to read; see the questions below. To guide the reader, it would also be beneficial if a brief description of the proof strategy is given before each proof.

Significance: This method is certainly interesting and probably useful in the context of Bayesian inference.

---

> ### Author Response · Authors · 2022-08-01
> **Response to Reviewer No1L Part 1: Response notation and Theorem 4.1 Questions**
>
> Thank you for your careful reading of our manuscript, and the helpful suggestions you have made. We endeavour to respond to all the comments here - and have also submitted a revised version of our manuscript and the Supplementary material. To avoid confusion, throughout our responses we will state when we are referring to line numbers in the (original) Supplementary material (OLD-SUP), or in the Revised Supplementary material (NEW-SUP).
>
> ## Theorem 4.1
>
> **Statement:**
>
> > I do not find this explanation very clear... The limiting behaviour for $J(\delta$  at line 189 is not obvious...
>
> Thank you for pointing out that this needs clarification. We have moved this example to Lines 24-43 of the NEW-SUP, where we precisely define the relevant terms that we introduce, and provide a full derivation of the the limiting behaviour of $J(\delta)$ in this setting.
>
> **Roadmap:**
>
> > Could you give a roadmap for this proof?...
>
> This is a very good idea – thank you for suggesting this. We have added roadmaps to the proofs of Theorems 4.1 and 4.2. These are given at the start of each proof, on Lines 71-87 and 139-178 of the NEW-SUP respectively.
>
> **Line 61:**
>
> > Can you give a few words of explanation? Why is this integral equal to the KL divergence?
>
> The variable $t$ is indeed in $[0,1]$, we will make sure to clarify this when it is introduced.
>
> The equation on Line 62 of the OLD-SUP follows from the form for the KL divergence derived by [1]. Thank you for pointing out that this was not clear in the proof. We give the full derivation of this equation on Lines 98-113 of the NEW-SUP. We also note that there is a slight typo in original version of this equation found in the OLD-SUP. We have corrected this in the NEW-SUP (the new relevant equation is Equation (8) of the NEW-SUP). All the subsequent results use the correct form of this result (in particular, see Lines 188-189 of the NEW-SUP), and so are unaffected by this change.
>
> **Line 75:**
>
> > the authors claim they use a union bound. This step necessitates a clarification...
>
> The union bound is over the events $\langle a, f_1- \bar f \rangle  \leq 2 t_N$ and $\|f_1 - \bar f \| > t_N/\phi$. This is because, if these two inequalities hold, then necessarily $  \langle a, f_1- \bar f\rangle - \phi\|f_1 - \bar f \| \leq  t_N $ by the triangle inequality. We do not bound $\phi$, instead, we choose its value on Line 78 of the OLD-SUP (Line 131 of the NEW-SUP).
>
> **Line 79:**
>
> > At line 79 (SM), there is a typo...
>
> Thank you for spotting this typo, we have corrected it in the revision.
>
> **Line 82:**
>
> > At line 82 (SM), I presume there is another typo...
>
> This is indeed a typo, the last line should be an inequality, and $\log(A_d^{-d/2})$ should be replaced by $\log(A_d^{2/d})$. Furthermore, the condition on $C_1$ should be: $C_1 \leq \log (A_d^{2/d})$.
>
> ## Appendix
>
> [1] Campbell, T. and Beronov, B., 2019. Sparse variational inference: Bayesian coresets from scratch. Advances in Neural Information Processing Systems, 32.

---

> > ### Comment · Reviewer_No1L · 2022-08-06
> > **General response**
> >
> > I thank the authors for providing clarifications and for writing a revised version of the manuscript and the SM.
> > This provided answers to my questions.
> > Consequently, I will increase my score.

---

> > > ### Author Response · Authors · 2022-08-08
> > > **Re: General response**
> > >
> > > Thank you very much for your reply, and evaluating our revised manuscript. We are pleased to hear that the changes helped provide answers to your questions.

---

> ### Author Response · Authors · 2022-08-01
> **Response to Reviewer No1L Part 2: Theorem 4.2 Questions**
>
> ## Theorem 4.2
>
> **Line 98:**
>
> > At line 98 (SM), the authors upper bound $I_{\epsilon,1}$. The last inequality is unclear...
>
> This is a mistake in the proof, thank you for picking this up. We have corrected it in the NEW-SUP, and give a sketch of this fix here. The idea behind this inequality was to upper bound a variance term by a mean-squared term. However, it was not correct to do this in the integral $I_{\epsilon,1}$; it needs to be done earlier on the integral over $\Theta$. In order to make sure that the bounds on both $I_{\epsilon,1}$ and $I_{\epsilon,2}$ hold, we also need to make the substitution found on Line 145 of the OLD-SUP earlier on. We do this on Line 182 of the NEW-SUP.  Next, we use the bound on the variance to form the inequality on Line 184 of the NEW-SUP. The bound on $I_{\epsilon,2}$ now follows through as before, except for a minor modification found on Lines 244-246 of the NEW-SUP.
>
> The bound on $I_{\epsilon,1}$ also follows very similarly to before. In essence, Line 98 of the OLD-SUP has been replaced by Line 201 of the NEW-SUP. This change is carried through into the final upper bound for $I_{\epsilon,1}$ found on Line 238 of the NEW-SUP. Throughout, the proof technique is exactly the same, except that we have replaced the $f_n$ by $\Delta_n \coloneqq f_n(\theta) - f_n(\theta_0)$ in several places. This leads to a modification of the final bound on the KL on line 264 of the NEW-SUP: $f_1 - \bar f$ is replaced by $\Delta_1 - \bar \Delta$, where $\bar{\Delta} \coloneqq \mathbb{E}_p(\Delta_n)$. However, the argument remains the same, and the proof hold as before. The asymptotics are also the same, it is just the constant that has changed.
>
> **Line 101:**
>
> > To be complete, it should be mentioned that $N$ has to be large enough...
>
> Thank you for this note, we will add this.
>
> **Line 100:**
>
> > This is not an assumption since the authors calculate right after the probability that this event does not happen...
>
> Thank you, we will fix this to clarify that we are working on the event that these inequalities hold. We then calculate the probability of the relevant events occurring, and these probabilities are included in the final high-probability bounds.
>
> **Line 102:**
>
> > At line 102 (SM), another inequality is unclear...
>
> Thank you for pointing this out, the inequalities should actually be for the norm on the left hand side, rather than the norm squared. Each line follows through as before, without the square. This tells us that
>
> \begin{align}
> \left\| \left\|(\theta - \theta_0) - \frac{S_N(\theta_0)}{2\sqrt{N}L_0} \right\| \right\| \geq \frac{1}{\sqrt{2}} \| \|\theta - \theta_0\| \|
> \end{align}
>
> On $B_{\epsilon}^c$. The required bound can now be found by squaring both sides, as they are positive numbers. We have fixed this in the revision.
>
> **Line 127:**
>
> > At line 127 (SM), the first equality is unclear...
>
> The equality follows from the fact that the left hand terms on the first and second line of 127 (in the OLD-SUP, Line 204 in the NEW-SUP) sum to give twice the right hand term. Thus if one is less than that term, the other must be greater than it. We will clarify this.

---

> ### Author Response · Authors · 2022-08-01
> **Response to Reviewer No1L Part 3: Theorem 4.3 Questions and Typos**
>
> ## Theorem 4.3
>
> **Line 229:**
>
> > At line 229, I think that the assumptions are incomplete...
>
> Thank you for pointing out that our conditions were not clear. We do indeed require that $1 - \gamma \xi \geq 0$, as we can see on Line 290 of the NEW-SUP. However, the condition that $\gamma \in [0,1]$ given on Line 229 of the main text, along with the fact that $\xi \leq 1 $ by definition, means that this condition is always satisfied. This, combined with the fact that $\epsilon \geq 0$ means that $\eta \geq 0$. It is true that $\eta$ can be $\geq 1$, in which case our result still holds, but is vacuous. However, we show that there is at least one broad set of cases where the result does give a useful bound (namely when the assumptions of Theorem 4.1 hold, and we can take $\epsilon = \delta = 0$). We will clarify this.
>
> **Global Optimum:**
>
> > What does this bound tell about the convergence to a global optimum?
>
> By definition, $W^*$ is a subset of the set of globally optimal points. Thus, $w_k^*$ is globally optimal $\forall k$, and our bound gives the rate of convergence to the global optimum. We will clarify this.
>
> ## Typos
>
> Thank you for spotting these typos, they will all be corrected.

---

### Official Review · Reviewer_nwJJ · 2022-07-07

**Rating:** 7
**Confidence:** 3
**Soundness:** 4 excellent
**Presentation:** 3 good
**Contribution:** 3 good

**Summary:**

This paper addresses the problem of approximating a Bayesian posterior with a large number of datapoints. They do so by selecting a sparse, weighted subset of datapoints (a coreset). Previous methods have addressed this problem, but the authors note that previous methods suffer from requiring some subset of: (1) potentially expensive loops over the whole dataset, (2) user input or parameter tuning, (3) a large number of potentially expensive MCMC simulations. The authors address (1) by noting that a coreset can be formed on a random subset of the data, (2) by proposing a quasi-Newton method that requires one MCMC simulation per iteration, and (3) just by virtue of providing a different algorithm without the same tuning / user input requirements.

The authors prove conditions under which (1) still allows for exact or approximate recovery of the posterior. And they prove conditions under which their optimization algorithm (2) does not require many iterations. In experiments, the authors show that their method is both faster and more accurate than other coreset methods.

**Questions:**

- Can the authors clarify what's going on with $\xi$ in Theorem 4.3 (see weaknesses above)?

- (minor question): On lines 118-122, the authors note that when M is larger than the dimension of the space of log-likelihood functions, $G$ is not invertible, which motivates adding some small $\tau$ to its diagonal. If $G$ has a zero eigenvalue in some directions, this implies the weight vector $w$ has an orthogonal decomposition into directions along which its variation does and does not matter. Could a reparameterization allow the optimization algorithm to instead just work along the directions for which the variation of $w$ does matter? This might avoid the use of $\tau$.

**Limitations:**

I think the authors have adequately discussed the social impact of their work.

**Strengths And Weaknesses:**

**Strengths**
- The writing in the paper is good; the material is pretty technical, but I found it relatively easy to follow.
- The method proposed by the authors intuitively fixes clearly explained problems with previous work and has desirable theoretical properties (mostly -- see weaknesses below) and this intuition and theory actually seem to hold up in practice in non-exhaustive, but at least detailed and careful experiments. That's a pretty solid combination. Based on this, I vote for the paper to be accepted.

**Weaknesses**
1. The method is claimed to be "black box" with "far less ... user input required" than other algorithms. I think this is overselling things a little bit. I see five different hyperparameters (at least five, since one of them is a sequence): the number of coreset points $M$, the step size sequence $\gamma_k$, the Hessian regularization $\tau$ (or, equivalently, the condition number threshold mentioned on line 142), the number of iterations to tune the stepsize for $K_{tune}$, and the size of the subsampling set $T$. Since there are no sensitivity studies for these hyperparameters, I don't think it's fair to suggest that little user input is required. Relatedly, some competing methods are criticized for needing an initial, user-specified posterior approximation $\hat\pi$ as an input, such as the Laplace approximation. I don't think this requires much user input; given access to an optimization library and automatic differentiation tools, a Laplace approximation can be constructed without user input.
2. Theorem 4.3, which studies the convergence rate of the authors' proposed algorithm (an important selling point of the work), essentially assumes the smallest eigenvalue of the covariance $G(w)$ is not too small. For example if $\xi$, is zero, and the optimal coreset perfectly approximates the posterior, then Theorem 4.3 predicts no convergence of their algorithm. But it seems -- and maybe the authors can correct me on this -- that as $N \to \infty$ we should expect $\xi \to 0$. For example, say we expect the posterior to concentrate to a point mass as $N \to \infty$ and some $w^* \in W$ perfectly approximates the posterior. If the posterior is contracting to a point mass, won't $G(w^*)$ approach the matrix of 0's? (I'm writing this as if the inf over $W$ is actually the inf over $W_N$; if it's the inf over $W$, then the inf contains arbitrarily large $w$'s even for $N = 1$).

**Smaller things**

- There are a lot of derivatives with respect to $w$ taken throughout the paper. But $w$ sits in a constrained set $W$. How are these derivatives defined when $w$ is on the boundary of $W$?
- Should the left hand side of equations 6 and 7 be indexed $1:M$ and $1:M, 1:M$, respectively?
- Line 113: it's not intuitively clear to me why the first term should dominate the expression here.
- Algorithm 1: is input $K$ supposed to be $K_{tune}$?
- "This is sublinear in $N$". I usually take "sublinear" to mean $o(N)$, whereas this runtime is $O(N)$.
- In the equation right after line 188, I don't think $A$, $K, X$, and $X_1$ were ever defined.
- In Theorem 4.3: could the inf over $w \in W$ be replaced by the inf over ${ w \in W :  \|\| w - w^* \|\| \leq \|\|w_0 - w^* \|\| \}$? Since there's monotone improvement towards $w^*$, it seems like you'll never leave this ball, and so you can ignore any bad behavior outside of it.
- In the experiments, it would be good to include the runtime of FULL to see how marginal / non-marginal the gains here are.
- In the proof of Theorem 4.3, there are a couple places where $G(w) + \lambda I$ is written. Should this $\lambda$ be a $\tau$?
- It would be helpful if the equations in the appendix were contiguously numbered with those in the main text. Otherwise it's not immediately obvious which equation "Eq. (1)" refers to.
- The letter $t$ shows up in a few places in proofs. It seems like this is a number between 0 and 1, but I don't think this is ever stated.

---

> ### Author Response · Authors · 2022-08-01
> **Response to Reviewer nwJJ Part 1: Response notation and Weaknesses**
>
> Thank you for your careful reading of our manuscript, and the helpful suggestions you have made. We endeavour to respond to all the comments here - and have also submitted a revised version of our manuscript and the Supplementary material. To avoid confusion, throughout our responses we will state when we are referring to line numbers in the (original) Supplementary material (OLD-SUP), or in the Revised Supplementary material (NEW-SUP).
>
> # Weaknesses
>
> ## Experiments
>
> ### Sensitivity Analyses
>
> > Since there are no sensitivity studies for these hyperparameters...
>
> Thank you for this suggestion. We agree that given the set of tuning parameters you identified, we should have included a sensitivity analysis in the original version. We have now added a sensitivity analysis for $S$, $K_{tune}$ and $\tau$ to the NEW-SUP. We discuss these in the response to Reviewer LhtQ, please refer to that response for further details: https://openreview.net/forum?id=M3WW7TqoMvc&noteId=wcBj6N_v3y.
>
> With regards to subsampled set $T$, this remark was added as a possibility to reduce computation that requires deeper study. We should have been clearer that this was the intention, and we will clarify this. We do not use it in our experiments.
>
>
>
> ### Other methods
>
> > Relatedly, some competing methods are criticized for needing an initial, user-specified posterior approximation...
>
> It is correct to say that obtaining a Laplace approximation to use as $\hat{\pi}$ can be done without much user input. However, the problem here is that this choice of $\hat{\pi}$ does not lead to good results, as we see in Section C.5 of the OLD-SUP. We will clarify that the problem we mention here is that it is difficult to find a **good** choice of $\hat{\pi}$ without significant user input. As we discuss briefly in Section C.6, these methods are further limited by using a fixed approximation, no matter its quality.
>
> ## Theory
>
> ### Theorem 4.3
>
> > But it seems -- and maybe the authors can correct me on this -- that as $N \to \infty$ we should expect $\xi \to 0$ ...
>
> This is a really fantastic question, thank you for raising it. We will add some clarification to the manuscript. If $\xi = 0$ then there is indeed no convergence. However, it is not true that as $N \to \infty$ we necessarily have that $\xi \to 0$. This is essentially because, for any fixed $N$, we are free to choose $\tau$. So, while $G(w)$ does shrink as $N \to \infty$, we can (and should) choose $\tau$ to be sufficiently small that $\xi$ is roughly $1$. In particular, this means setting $\tau$ significantly smaller than the minimum positive eigenvalue. This strategy does assume that, for each fixed $N$, the eigenvalues are uniformly bounded away from zero by some amount, but it is OK for this amount to be shrinking with $N$.

---

> ### Author Response · Authors · 2022-08-01
> **Response to Reviewer nwJJ Part 2: Smaller Things and Questions**
>
> ## Smaller things
>
> **Derivatives:**
>
> >  How are these derivatives defined...
>
> Throughout, when we take derivatives we do this on the ambient space $\mathbb{R}^N$. Our algorithm involves a projection onto the feasible set $W$, and this is the standard approach in projected optimization methods.
>
> **Line 113:**
>
> > Line 113: it's not intuitively clear to me why the first term should dominate the expression here...
>
> Thank you for pointing out that this is unclear – we will clarify it. This intuition comes from the fact that the derivatives of $\log Z(w)$ can be expressed the centralised moments of $f(\theta)$. In particular,
> \begin{align}
>     \nabla_{w}^{3} \log Z(w)(1-w) = \mathbb{E}_{\pi_w} \left[ \left( f-\mathbb{E} _{ \pi_w } (f) \right) \left( f-\mathbb{E} _{ \pi_w } (f) \right)^T \left( (f^T (1-w) -\mathbb{E} _{ \pi_w } (f^T (1-w) ) \right) \right]
> \end{align}
>
> which is small when $w^Tf(\theta) \approx 1^Tf(\theta)$. Conversely, the second term in Line 113 does not contain this $f^T (1-w)$ term.
> However, we emphasise that this is motivational heuristic, which is subsequently rigorously justified by Theorem 4.3.
>
> **Sublinear in N:**
>
> > "This is sublinear in $N$" ...
>
> The sublinearity here is in the space complexity, not the time complexity (which is indeed linear in $N$). We will clarify this.
>
> **Line 188:**
>
> >  I don't think $A$, $K$, $X$ and $X_1$ were ever defined.
>
> We have moved this example to Lines 24-43 of the NEW-SUP, where we precisely define these terms, and provide a more full treatment of the intuition behind the limiting behaviour of $J(\delta)$ in this setting.
>
> **FULL Runtime:**
>
> > In the experiments, it would be good to include the runtime of FULL ...
>
> Thank you for this suggestion, and we will add some discussion on this topic. In Section C of the OLD-SUP, we see the per-iteration time taken to sample from the full posterior, as well as the respective coreset posteriors.
>
> In order to assess the computation gains our coreset approach achieves, we believe that a useful metric is to calculate the number of posterior samples $N_{sample}$ for which the time taken to obtain $N_{sample}$ samples from the full posterior is the same as the time taken to construct a coreset, and then take $N_{sample}$ from the coreset posterior. We estimate this value for the sparse linear and logistic experiments we find values of $N_{sample} \approx 60$ and $N_{sample} \approx 120$ respectively. This is far fewer than you would ideally like to have in practice.
>
> However, we note that these results are conservative, and in fact we expect that the gains from the coreset approach are more significant. This is because we perform sampling in each case using STAN, which uses C++. However, we construct the coresets in Python, and most of the coreset build time comes from the slowness of a non-compiled language. We expect that our estimated values of $N_{sample}$ in each case would be far smaller if we not only performed sampling in C++, but also constructed the coresets in C++.
>
> **Miscellanious:**
>
> All other typos and stylistic points will be addressed, thank you for pointing these out.
>
> ## Question
>
> > Can the authors clarify what's going on with $\xi$ ...
>
> Thank you for this question, we provide some intuition on this in https://openreview.net/forum?id=M3WW7TqoMvc&noteId=Ndwg8ig4ekX
>
> > Could a reparameterization allow the optimization algorithm to instead...
>
> In theory, the proposed reparametrization could be helpful, thank you for suggesting it. However, in practice it is difficult to distinguish between the exactly $0$ eigenvalues, and the ones with a very small positive value (due to floating point arithmetic). So, we use the standard approach of adding a diagonal regularization term.

---

> > ### Comment · Reviewer_nwJJ · 2022-08-07
> > **Reply**
> >
> > Thanks for the detailed replies, and thank you for updating the paper; that's really helpful. It seems like everyone is in agreement that the paper should be accepted.
> >
> > One thing about how the derivatives are defined: I should have been more clear that I think there's an issue with the exposition earlier on in the paper. For example, Eq (5) takes derivatives with respect to $w$, but right above that we have $w \in \mathbb{R}_{\geq 0}^N$. I think it would be fine to just say something like ", where derivatives with respect to $w$ are defined on the ambient space $\mathbb{R}^N".

---

> > > ### Author Response · Authors · 2022-08-08
> > > **Re: Reply**
> > >
> > > Thank you very much for your reply, and evaluating our revised manuscript.
> > >
> > > Thanks also for the clarification about the derivative definitions - we will add the note you suggest to ensure that this is clear in the paper.

---

### Official Review · Reviewer_hdj8 · 2022-07-08

**Rating:** 7
**Confidence:** 3
**Soundness:** 3 good
**Presentation:** 4 excellent
**Contribution:** 3 good

**Summary:**

In the area of Bayesian Coresets, this paper presents an algorithm that proposes:
1. To uniformly select $M$ data points that will be used to  approximate the posterior distribution, and,
1. to use a quasi-Newton method to refine the weights that will be assigned to each of the selected data points.

For this quasi-Newton method, the authors propose using an estimated covariance matrix to approximate the objective’s Hessian.
Finally, the authors provide theoretical justification for their algorithm via three different theorems.

**Questions:**

The most important thing I would like the authors to clarify is the derivation of equation (6) in the Appendix because it is critical for the proof of both, Thm. 4.1 and Thm. 4.2.

In the proof of Thm. 4.3, I think it should be that $\mathbf u \in \mathbb R^M$ is in the null space of $G(\mathbf w)$ iff $\mathbf{u}^{\mathrm{T}}f(\theta) = c$ holds   $\pi_w$-almost everywhere.

For the following limitations, I'm not asking the authors to solve them. Instead, I suggest the authors that, if my assessment is correct, they should make the limitations more explicit as open problems.

First, I see the following limitations:

- The use of Monte Carlo estimations.
This has two implications:
    1. The obvious one is the Monte-Carlo error, which might imply a bias.
    2. A not-so-obvious one is that the method assumes that we have an oracle that allows us to sample from the corresponding distribution. However, often times we can only get approximated samples from the distribution (or samples from an unconverged chain).

Part of point (1) was briefly mentioned, but there is no discussion about (2). To be concrete, it should be pointed out that a version of Thm. 4.3 with the two sources of errors, is still an open problem, extending the discussion at the end of section 3.2

Another limitation is the fact that we don't know if the optimization is actually making progress. A quick look at Algorithm 1 will show that there is no function of the objective being used as stopping criteria. I understand why this happens. Nevertheless, I think the authors should be, again, more explicit about this. This is related to the fact that the Armijo condition is not being used.

**Limitations:**

The authors mention some of the paper’s limitations. Above I'm suggesting mentioning a few more.

**Strengths And Weaknesses:**

I think this is a very solid submission.
The authors propose an algorithm and prove that their approximation might behave as the full posterior and provide a convergence rate.
A notable feature is that assumptions are deeply discussed in the body of the paper (Section 4) or the appendix.
In general, I would say that it is a very precise paper.
On the weak side, I found the proof in the appendix quite hard to follow. Many inequalities were not evident at all and limited the possibility of doing a complete verification of the paper’s contributions [cf. Questions below].
It is important to highlight that when I was able to derive the bounds, the proofs are correct, as in Thm. 4.3.

---

> ### Author Response · Authors · 2022-08-01
> **Response to Reviewer hdj8 Part 1: Response notation and Proofs**
>
> Thank you for your careful reading of our manuscript, and the helpful suggestions you have made. We endeavour to respond to all the comments here - and have also submitted a revised version of our manuscript and the Supplementary material. To avoid confusion, throughout our responses we will state when we are referring to line numbers in the (original) Supplementary material (OLD-SUP), or in the Revised Supplementary material (NEW-SUP).
>
> **Clarification on Equation (6):**
>
> This equation follows from the form for the KL divergence derived by [1]. Thank you for pointing out that this was not clear in the proof. We give the full derivation on Lines 98-113 of the NEW-SUP. We also note that there is a slight typo in version of Equation (6) in the OLD-SUP. We have corrected this in the NEW-SUP (the new relevant equation is Equation (8) of the NEW-SUP). All the subsequent results use the correct form of this result (in particular, see Lines 188-189), and so are unaffected by this change.
>
> We should note here that, whilst this equation is critical to the proof of Theorem 4.2, we do not need it for the proof of Theorem 4.1, since if Equation (5) of the OLD-SUP holds, then $\pi_w=\pi$ and thus trivially $\mathrm{KL}(\pi_{w(v)}||\pi) =0$.
>
>
> **Theorem 4.3 Proof:**
>
> Thank you for catching this – we will fix it.
>
> ### Appendix
>
> [1] Campbell, T. and Beronov, B., 2019. Sparse variational inference: Bayesian coresets from scratch. Advances in Neural Information Processing Systems, 32.

---

> > ### Comment · Reviewer_hdj8 · 2022-08-08
> > **Notation and Proofs**
> >
> > I appreciate the authors' response and the inclusion of a "Roadmap" for the Proofs of Thms 4.1 and 4.2. I think that improved the readability of the proofs.
> >
> > Small typo, the pdf of Beta(1, 2) is actually $f_T(t) = 2(1-t)$.
> >
> > As a subjective suggestion, I think both proofs could be improved using small Lemmas. Especially when using bounds of inner product spaces, which can be proven elementarily.

---

> > > ### Author Response · Authors · 2022-08-08
> > > **Re: Notation and Proofs**
> > >
> > > Thank you very much for your reply. We are pleased to hear that the roadmaps helped improve the readability proof, but thanks also for the suggestion of splitting the proof up into smaller lemmas. We will definitely consider this when thinking how to further improve the proof.
> > >
> > > Thanks also for picking up this additional typo, which we will fix.

---

> ### Author Response · Authors · 2022-08-01
> **Response to Reviewer hdj8 Part 2: Limitations**
>
> **Limitations:**
>
> Thank you for pointing out these additional limitations, in both cases we will add discussion to our manuscript in order to make these explicit.
>
> With regards to the Monte Carlo errors, we agree that this second type of error needs to be discussed, and we will add this. Furthermore, we should have clarified that a version of Theorem 4.3 that accounts for these sources of error is beyond the scope of our current work, and is an open problem. We will add this discussion to Section 3.2, as suggested.
>
> With regards to the optimization, again we agree that we need to be clearer about not being able to use the objective for monitoring. In our algorithm we use the norm of the gradient as to monitor progress, and stop early if there is not a significant decrease. We will add further discussion to clarify both of these points.

---

### Official Review · Reviewer_LhtQ · 2022-07-08

**Rating:** 8
**Confidence:** 3
**Soundness:** 4 excellent
**Presentation:** 4 excellent
**Contribution:** 3 good

**Summary:**

This paper presents a novel algorithm for approximating posterior distributions via weighted subsets of the data. The proposed algorithm samples points uniformly at random and then computes the weights using a quasi-Newton method; crucially it scales well with respect to the number of data points, both in terms of time and storage. The authors provide a rigorous theoretical analysis of their proposed algorithm and empirically evaluate it against commonly-used methods for posterior approximation.

**Questions:**

My understanding of the empirical evaluation is that all four settings satisfy the assumptions of either Theorem 4.1 or 4.2: is this correct and if so, have the authors considered experimenting on settings where those assumptions do not hold? It would be illuminating to see the empirical performance when the theoretical guarantees do not apply.

**Limitations:**

Yes, I believe the authors' discussion of their work's limitations is sufficient; related to my question above, I would have found it helpful if the authors had provided some intuition about settings in which their theoretical results would not hold much like how they did for settings in which the assumptions do hold.

**Strengths And Weaknesses:**

Overall, I found this paper to be an excellent read and believe it to be a strong candidate for acceptance.

1. The proposed method is not particularly creative but frankly, I think the simplicity of the ideas being presented here are a strength of this work. The sum of the contributions in my mind are sufficient to get this paper above the originality bar for acceptance.

2. The theoretical and empirical analysis are compelling although I do have a few suggestions/concerns regarding the experiments that if addressed, could raise my score of this submission even higher:
    - I would have liked to see some comparison against SVI, even if that meant scaling down the experiments to settings where SVI is computationally tractable. It would be very telling if QNC was competitive with or even outperformed SVI in small-data regimes; if not, then finding some sort of cost-accuracy trade-off between the two methods could be useful for practitioners.
   - I would have liked to see some sort of sensitivity analysis to the model hyperparameters, specifically S, K_tune (about which very little is said) and tau.

3. The manuscript is very well-written; I genuinely found this paper to be an enjoyable read and applaud the authors' effort to include intuition wherever possible.

4. I believe this work will have significant impact on a non-trivial percentage of the NeurIPS population: the proposed algorithm is simple and easy-to-implement and thus, accessible to people who might want to use/expand upon it. Furthermore, the scalability makes this algorithm broadly applicable, in settings where previously-proposed methods might not be.

---

> ### Author Response · Authors · 2022-08-01
> **Response to Reviewer LhtQ Part 1: Response notation and Experiments**
>
> Thank you for your careful reading of our manuscript, and the helpful suggestions you have made. We endeavour to respond to all the comments here - and have also submitted a revised version of our manuscript and the Supplementary material. To avoid confusion, throughout our responses we will state when we are referring to line numbers in the (original) Supplementary material (OLD-SUP), or in the Revised Supplementary material (NEW-SUP).
>
> ## Experiments
>
> ### SVI Comparison
>
> We have revised our manuscript, and in Section C.7 of the NEW-SUP we provide a comparison against SVI on a smaller dataset. This experiment is the same as that in Section 5.1, except that we have reduced the dimension from $100$ to $50$, and the dataset size from $1,000,000$ to $10,000$. For SVI we use $100$ optimization iterations, and only run 1 trial due to time constraints. We see that, even in this smaller data setting, the performance of SVI is not comparable. In reality, the number of optimization iterations needed is much higher than $100$, which is reflected in the poor performance of SVI. But, we see that even for this number the time to build a coreset of a size for which our method performs well is several orders of magnitude slower than any other method. The cost-accuracy trade off is therefore heavily weighted in favour of our model, but we agree that investigating this further in the small-data regime could be a useful exercise.
>
> ### Sensitivity Analyses
>
> Thank you for this suggestion – we have added a sensitivity analysis for $S$, $K_{tune}$ and $\tau$ to Section C.8 of the NEW-SUP. We do this analysis by repeating the Bayesian sparse linear regression experiment detailed in Section 5.2 for varying values of one of these parameters at a time (with all other parameters kept fixed). Due to time constraints in the rebuttal period, we only perform 1 trial here in each case. However, we will complete these experiments so that we may plot the $25^{th}/75^{th}$ percentiles over 10 trials as in the original experiments. We summarise the results of these initial experiments here.
>
> $\textbf{S:}$
>
> In our original experiments, we use $S=500$ throughout. From this analysis, we see that the performance of our method holds up for substantially lower values of $S$ (though taking $S=10$ is too low and leads to a degradation in performance). Moreover, doubling the number of samples does not lead to significantly better performance. The build time generally increases for larger $S$, as we would expect.
>
> $\mathbf{ \tau :}$
>
> In our original experiments, we use $\tau = 0.01$ throughout. From this analysis, we see that taking too large a value of $\tau$ negatively affects the performance. This is in line with what we expect, since larger values skew the step away from a true Newton step. Our theory recommends taking $\tau$ as small as possible such that $G(w) + \tau I$ is still (numerically) invertible. Indeed, we see that decreasing $\tau$ can improve the performance of our model, though our choice still provides good results. Thus, we believe ours is a conservative choice that works well in practice.
>
>
> $\mathbf{K_{tune}:}$
>
> In our original experiments, we use $K_{tune} = 1$ throughout. In this analysis, we see that the choice of this parameter has very little effect on the performance of our method for this experiment. In our original experiments, we take $\gamma_k = 1$ for $k > K_{tune}$. When we perform our line search, our starting value is also $\gamma_k = 1$. We find that the line search stops immediately, meaning that for every value of $K_{tune}$ we get essentially the same results.
>
> The purpose of this step in our algorithm is to guard against the case where our initial gradient and covariance estimates are very noisy, and we may want to take a smaller step initially. However, we see that this is in fact not needed for this experiment.

---

> > ### Comment · Reviewer_LhtQ · 2022-08-08
> > **Acknowledgment of response**
> >
> > I thank the authors for their detailed response and for the extra work they put into performing the ablation study. I find the new results very compelling and am thus, increasing my score to an 8.

---

> > > ### Author Response · Authors · 2022-08-08
> > > **Re: Acknowledgment of response**
> > >
> > > Thank you very much for your reply, and considering our new results. We are pleased to hear that you found these convincing - thank you for helping us to improve our work by including them.

---

> ### Author Response · Authors · 2022-08-01
> **Response to Reviewer LhtQ Part 2: Questions and Limitations**
>
> ## Questions
>
> ### Theoretical Results
>
> Of our experiments, the Gaussian, sparse linear and radial basis function regression experiments all fall into the settings covered by our theory. However, the logistic regression experiment actually is not covered by the assumptions required for Theorem 4.2, since the log likelihood is concave but not strongly concave. It is therefore reassuring to see that our method obtains favourable empirical results in this experiment, despite our assumptions not holding precisely in this context.
>
> ## Limitations
>
> Thank you for suggesting this additional limitation, we agree that this would be helpful to add, and we will provide some intuition on this. Related to the above, it is the strong concavity condition which is not commonly satisfied in practice, such as in our logistic regression experiment.

---

### Meta-Review · Area_Chair_5ZZ8 · 2022-08-22

**Recommendation:** Accept
**Confidence:** Certain

**Metareview:**

The paper has genereated unanimous enthusiasm and we are happy to recommend acceptance. Please make sure that all comments in the reviews/discussion threads are taken into account in the final version of the manuscript.

**Award:**

No

---

### Decision · Program_Chairs · 2022-09-14

Accept